# The influence of government ideology on the rate of e-waste recycling in the European Union countries

Erdal Arslan[1], Cuneyt Koyuncu[2], Rasim Yilmaz[3]*

1 Department of Economics, Faculty of Economics and Administrative Sciences, Selçuk University, Konya, Turkey, 2 Department of Economics, Faculty of Economics and Administrative Sciences, Bilecik Seyh Edebali University, Bilecik, Turkey, 3 Department of Economics, Faculty of Economics and Administrative Sciences, Tekirdag Namik Kemal University, Tekirdag, Turkey

* rasimyilmaz@nku.edu.tr

**Data Availability Statement:** All relevant data are within the manuscript and its Supporting information files.

## Abstract

This study examines the impact of government ideology on e-waste recycling in 30 European countries from 2008 to 2018. This study seeks to enhance the e-waste recycling literature by introducing a novel determinant, examining the unexplored relationship between government ideology and e-waste recycling rates in European countries, thus addressing a substantial research gap. Utilizing panel quantile regression on an unbalanced dataset, the findings revealed that the increased influence of right-wing parties in government was associated with lower e-waste recycling rates. Conversely, greater influence of left-wing or center-wing parties was correlated with higher recycling rates across all quantiles analyzed. The estimation results remain robust when different indicators of government ideology were employed. Overall, the study underscores the importance of political ideology in shaping e-waste recycling policies and environmental sustainability efforts. It emphasizes that effective policies should align with the political commitment of the governing body.

## Introduction

Electronic waste, or e-waste, is comprised of discarded and secondary electrical and electronic products, which include a vast array of devices and equipment that we use in our daily lives. The amount of waste electrical and electronic equipment (WEEE or e-waste) generated worldwide is rapidly increasing and has significant environmental consequences. According to the Global E-waste Monitor 2020 report by Forti et al. [1], e-waste is one of the fastest-growing waste streams in the world, both in terms of volume and its impact on the environment.

E-waste contains both hazardous and precious materials. Hazardous materials of e-waste may lead to environmental problems as well as health problems due to the improper disposal of the e-waste. E-waste also includes useful and precious materials. That is why the collection, treatment, and recycling of e-waste with a reliable and proper e-waste management system is very important. This kind of e-waste management system reduces the amount of e-waste and hence decreases the environmental and health problems posed by e-waste. Also, recycling and

**Funding:** The authors received no specific funding for this work.

**Competing interests:** The authors have declared that no competing interests exist.

recovery of precious materials could reduce the demand for new raw materials and contribute to the economy. Thus, this kind of e-waste management system can enhance the circular economy and sustainable production and consumption, as well as raise resource efficiency.

The European Union (EU), recognizing the environmental challenges posed by e-waste, has demonstrated a steadfast commitment to addressing this issue. E-waste recycling and improved collection are key strategies for implementing the circular economy and the reduce-reuse-recycle approach advocated by the EU. The EU has established a robust regulatory framework to govern e-waste recycling. The Waste Electrical and Electronic Equipment (WEEE) Directive, introduced in 2003 and revised in 2012, places responsibility on producers to manage the disposal and recycling of electronic products. This directive not only sets targets for the collection and recycling of e-waste but also encourages member states to implement measures that promote extended producer responsibility. The WEEE Directive implemented a gradual increase in collection targets starting from the reference years 2016 and 2019. The collection target started at 45% in 2016 and increased to 65% from 2019 onward [2].

The recycling rate of electronic waste (e-waste) exhibits notable variations among European countries, with significant disparities among member states. According to Eurostat (The Statistical Office of the European Union) data [3], the European Union achieved an overall e-waste recycling rate of 38.9 percent in 2018. However, at the individual country level, recycling rates varied considerably, ranging from 83.4% in Croatia to 20.8% in Malta. Noteworthy member countries with high e-waste recycling rates include Denmark (67.5%), Bulgaria (66.7%), and the United Kingdom (67.0%). Conversely, countries with the lowest e-waste recycling rates include Italy (32.1%), Iceland (24.7%), and Romania (25.0%).

To attain the aforementioned goals of the EU, understanding the drivers that influence the e-waste recycling rate is crucial for developing effective policies and strategies to mitigate the environmental impact of e-waste. As posited by Cadoret and Padovano [4], the considerable disparities in e-waste recycling rates within a cluster of closely connected and largely homogeneous EU economies give rise to concerns about the efficacy of models that rely solely on economic, social, and environmental factors. The determination and impact of e-waste recycling rates in EU countries should also be notably shaped by political commitment and national-level factors. The objective of this study is to enrich the existing understanding of the factors that impact the e-waste recycling rate. It accomplishes this by exploring the impact of government ideology on the e-waste recycling rate across European countries.

The purpose of this article is to explore the influence of government ideology on the rate of e-waste recycling, as well as other related factors, in European countries. Previous studies have not considered government ideology as a factor affecting e-waste recycling. This study aims to fill this research gap by proposing a new determinant that may affect e-waste recycling rates. To the best of our knowledge, no empirical research has examined the impact of government ideology on e-waste recycling rates. This research contributes to the existing literature by identifying government ideology as a new factor affecting e-waste recycling rates. Additionally, the study sheds light on the impact of government ideology on the environment by providing new evidence of its influence on environmental improvement, as represented by the e-waste recycling rate.

The remaining part of this study is structured as follows: The Literature Review section offers a review of relevant literature. In Research Hypothesis section, we outline the primary theoretical arguments connecting government ideology with e-waste recycling rate. The Research Methodology and Data section explains the research methodology and data used in the study. The estimation method employed in the analysis is discussed in the Estimation Technique section. The Estimation Results section presents the empirical results, and in the Discussion Section, we engage in a discussion on the empirical findings. Finally, the concluding section summarizes the key findings and provides policy implications.

## Literature review

The literature review section of this study is divided into two parts. The first part reviews the existing literature on the determinants of e-waste recycling, while the second part examines the results of empirical studies that have investigated the impact of government ideology on the environment.

### Literature review on the determinants of e-waste recycling

The empirical literature on e-waste recycling is a subset of the broader literature on recycling. Specifically, studies on the determinants of e-waste recycling can be categorized into two main groups.

The first group of studies primarily employ household-level data, often collected through surveys, to identify individual-level and micro-level factors that influence e-waste recycling behaviors. Several such studies have investigated the determinants of e-waste recycling decisions in different countries (see, for example, [5–7]). The literature focusing on household-level data has revealed a multitude of factors that influence individuals' recycling behavior. These factors include access to recycling, age, attitude towards recycling, education level, employment status, environmental awareness, environmental norms and beliefs, gender, household size, income, institutional support, laws and regulations, marital status, publicity, recycling costs, and recycling convenience.

The second group of studies examines aggregate data and focuses on macro-level factors that influence the rate of e-waste recycling. Cross-country studies on the determinants of e-waste recycling are relatively scarce. Boubellouta and Kusch-Brand [8] aimed to identify the driving factors of e-waste recycling rate in 30 European countries by utilizing data over the 2008–2018 period. According to the study's results, economic growth and e-waste collection were found to be the main drivers of the e-waste recycling rate, while population, energy intensity, and credit to the private sector also had an impact on the e-waste recycling rate. Yilmaz and Koyuncu [9] investigated the impact of globalization and its three sub-components on the e-waste recycling rate in European nations. Their findings indicated that, globalization and its three sub-components exert a favorable influence on the e-waste recycling rates in 30 European countries. Meanwhile, Constantinescu et al. [10] examined the effect of eco-investment on e-waste recycling in 24 EU countries from 2009 to 2018 and found that eco-investment per capita had a positive impact on e-waste recycled per capita during the estimation period.

This study aims to contribute to the literature on e-waste recycling by proposing a new determinant that has not been previously explored in empirical research. Specifically, the study investigates the impact of government ideology on the rate of e-waste recycling in European countries. To the best of our knowledge, no previous empirical research has examined this relationship. Therefore, this study fills a significant gap in the literature by identifying government ideology as a potential determinant of the rate of e-waste recycling.

### The impact of government ideology on environment

This section of the study provides an overview of cross-country research investigating the influence of government ideology on the environment. When reviewing empirical studies on the impact of government ideology on the environment, it becomes apparent that a majority of these studies employ proxies such as greenhouse gas emissions, environmental quality, the Environmental Performance Index, and climate policy to measure environmental impacts.

Examining the relationship between air pollution and the strength of left-wing parties in seventeen OECD countries, King and Borchardt [11] discovered a moderate yet sustainable inverse correlation between left party strength and per capita levels of air pollution. In her study, McKitrick [12] analyzed the impact of the ruling party on urban air pollution in 13 Canadian cities and concluded that both right-wing and left-wing provincial parties are linked to increased levels of certain air contaminants. Utilizing sulfur dioxide ($SO_2$) concentration data for 107 cities in 42 countries spanning from 1971 to 1996, Bernauer and Koubi [13] found a negative relationship between pollution and the strength of green parties. Garmann [14] examined the effects of government ideology on the process of reducing carbon dioxide emissions in 19 OECD countries from 1992 to 2008. The study's findings indicate that center and left-wing governments are associated with greater reductions in emissions compared to right-wing governments. Using a balanced panel of 65 countries spanning from 1981 to 2012, Chang et al. [15] determined that left-wing governments are correlated with lower carbon dioxide emissions in the least polluted countries. However, the study did not find significant influences of government ideology on carbon dioxide emissions in the median and most polluted nations. Jahn [16] examined the impact of party families in government on greenhouse gas emissions (GHGEs) across 21 highly industrialized countries. The study suggests that left parties are more effective in reducing GHGEs, the inclusion of green parties in government leads to decreased GHGEs, and the presence of non-Christian center parties has a negative effect on GHGEs.

Using panel data encompassing 85 countries from 2002 to 2012, Wen et al. [17] examined the relationship between government ideology and environmental quality. The estimation results of their study reveal that left-wing governments prioritize environmental quality over economic performance, whereas right-wing governments place greater emphasis on economic growth rather than environmental concerns. However, their findings also suggest that both left- and right-wing governments tend to compromise environmental objectives for higher economic growth when faced with voter pressure for improved economic performance. Using the Environmental Performance Index (EPI), a comprehensive measure of environmental quality, Kammerlander and Schulzetest [18] examined whether left-wing governments demonstrate a greater concern for the environment. Their study focuses on a panel of 134 countries spanning from 2007 to 2016. The empirical estimation results of their analysis indicated that left-wing governments outperform right-wing governments in terms of environmental performance. Centrist governments exhibited the highest EPI scores among the political orientations considered.

Cetkovic and Hagemann [19] conducted a study examining the influence of populist radical-right parties (PRRPs) on energy and climate policy in six West European countries: Austria, Denmark, Finland, the Netherlands, Norway, and Sweden. The analysis covers the period from 2008 to 2018. The findings of their analysis demonstrated that energy and climate policy initiatives can experience a decline when PRRPs participate in government or lend support to minority governments. In an analysis of twenty-nine democracies from 1990 to 2016, Schulze [20] found that left governments implement more stringent climate policies compared to center and right governments. This effect is particularly pronounced leading up to elections. The study concludes that partisan and electoral incentives play a crucial role in addressing climate change effectively.

Based on our analysis of the literature, we have found that government ideology has a significant influence on various environmental indicators. This study also contributes to the existing body of research on the relationship between government ideology and the environment. It presents novel evidence supporting the effect of government ideology on environmental progress, specifically in terms of e-waste recycling rates.

## Research hypothesis

The relationship between political factors and the environment has been a subject of extensive research and debate. Two prominent theoretical frameworks, the median voter model (also known as the consensus hypothesis) and the partisan theory, offer predictions regarding the impact of political factors on environmental policy.

The median voter model and the consensus hypothesis propose that political parties tend to converge towards the preferences of the median voter in order to gain electoral support, leading to policy alignment with the median voter's position. Within this framework, environmental issues are often considered nonpartisan or valence issues, as they impact the lives of individuals regardless of their political affiliations. Consequently, the consensus hypothesis suggests that partisan differences among political parties do not significantly affect environmental issues. Therefore, a testable hypothesis derived from the consensus hypothesis regarding the rate of e-waste recycling could be that there are no discernible differences among governments in their approaches to the rate of e-waste recycling [14, 21, 22].

Hypothesis 1: *There are no discernible differences among governments in their approaches to the rate of e-waste recycling.*

Conversely, the partisan theory argues that the policies and outcomes are shaped by the ideological positions of the political parties in power. According to this theory, political parties strive to implement policies that align with their own preferences and ideological leanings. These ideological inclinations guide political parties in advocating for policies that are consistent with their government's stance. Given that political parties hold diverse preferences and beliefs, their policy choices may have different economic implications. The variations in government ideology among political parties can lead to distinct attitudes towards policy and influence the likelihood of initiating reforms. Therefore, the ideological orientation of political parties plays a crucial role in elucidating the regulatory restructuring within a country [15–17, 23].

According to the partisan theory, the political context of a country, particularly the dominant political parties, holds substantial influence over the formulation of environmental protection policies and regulations. As political parties may possess divergent perspectives on environmental issues, significant variations can emerge. These disparities in government ideology have the potential to shape the development of environmental policy. Moreover, since nearly all environmental policies are crafted and executed within a political framework, the ideological stance of the ruling government can also impact the extent to which renewable energy is adopted. Consequently, the ideological orientation of political parties in power wields significant influence over the direction and implementation of environmental policies [15, 17, 23].

The partisan theory revolves around the ideological division between the left and the right political spectrums. The classical interpretation of this theory employs the left-right dichotomy as a foundational framework for analysis, classifying governing political parties as either the left or the right. This concept draws inspiration from earlier partisan theories, including those formulated by Hibbs [24] and Alesina [25]. These theories posit that left-wing parties prioritize policies that advance the interests and welfare of the working class, while right-wing parties prioritize policies focused on inflation control and fostering economic growth [16]. According to the partisan theory, there is a divergence in the treatment of environmental issues between left-wing and right-wing parties. Left-wing parties typically place greater emphasis on environmental concerns, environmental quality, and the protection of the environment compared to right-wing parties. As a result, left-wing parties tend to advocate for greater regulation and intervention in energy markets compared to their right-wing counterparts.

Government ideology, according to the partisan theory, plays a pivotal role in shaping e-waste recycling rates by determining the types and degrees of policies and initiatives aimed at managing electronic waste. It significantly influences e-waste recycling rates through the policies, regulations, and approaches adopted by governments of different ideological leanings. Consequently, the divergence in the treatment of environmental issues between left-wing and right-wing governments can markedly influence the approach toward managing electronic waste [16, 22].

Left-wing governments, aligning with their emphasis on environmental protection, tend to introduce and enforce regulations mandating e-waste recycling. These regulations often require manufacturers to assume responsibility for the end-of-life disposition of their products, thus promoting recycling and responsible disposal. Additionally, left-wing governments are more inclined to allocate funds and resources toward the development of e-waste recycling infrastructure, which could encompass establishing collection centers, investing in recycling technologies, and providing incentives for businesses to adopt sustainable practices. Furthermore, these governments often prioritize public awareness campaigns and educational programs designed to encourage e-waste recycling, funding initiatives that inform and incentivize citizens to recycle electronic waste, thereby increasing overall recycling rates. Moreover, left-leaning governments frequently advocate for Extended Producer Responsibility programs, compelling manufacturers to take responsibility for the entire lifecycle of their products, including disposal and recycling, significantly enhancing e-waste recycling rates [16, 22, 23, 26].

In contrast, right-wing governments may favor market-based solutions, incentivization, and voluntary industry-led initiatives over stringent regulations. Their approach might prioritize economic factors and market freedom rather than extensive government intervention, potentially not significantly enhancing e-waste recycling rates [4, 23]. Hence, a testable hypothesis derived from the partisan theory regarding the rate of e-waste recycling would suggest that there are discernible differences among governments in their approaches to the rate of e-waste recycling.

Hypothesis 2: *There are discernible differences among governments in their approaches to the rate of e-waste recycling.*

This study aims to examine how government ideology affects the rate of e-waste recycling by evaluating the validity of both the consensus hypothesis and partisan theory. The consensus hypothesis posits that there are no discernible differences among governments in their approaches to the rate of e-waste recycling. In contrast, the partisan theory argues that government ideology matters, with left-wing governments being more inclined to promote the rate of e-waste recycling compared to their right-wing counterparts. By analyzing a sample of 30 European countries (28 EU member countries, Norway, and Iceland), this study seeks to provide insights into the influence of government ideology on the rate of e-waste recycling.

## Research methodology and data

The IPAT (impact of population (P), affluence (A), and technology (T) on the environment (I) model introduced by Ehrlich and Holdren [27] and its stochastic form STIRPAT (stochastic impacts by regression on population, affluence and technology) model propounded by Dietz and Rosa [28] have been extensively utilized for empirical examination of the impacts of human being on the environment (e.g., habitat destruction, extinction of wildlife, destruction of ecosystems, river contamination, deforestation, air pollution etc.) in the literature. According to the IPAT and STIRPAT models, the impact of human being on the environment can be captured by three factors, namely population size (P, represented by total population),

affluence (A, mostly reflected by GDP per capita), and technology (T, mainly given by energy intensity): I = P x A x T.

Empirical studies often use total population and gross domestic product (GDP) per capita to proxy for Population and Affluence, respectively. Additionally, energy intensity is used as a proxy for Technology. To examine the potential non-linear relationship between Affluence and Environmental Degradation, as proposed by the Environmental Kuznetz Curve (EKC) hypothesis, GDP per capita square is also included in empirical studies (see, for example, [8]).

The utilization of the IPAT (Impact of Population, Affluence, and Technology on the Environment) and STIRPAT (Stochastic Impacts by Regression on Population, Affluence, and Technology) models in this study is justified by their well-established role in empirically examining the environmental impacts of human activities. In the context of this study, which investigates the influence of government ideology on the e-waste recycling rate in European countries, the integration of the government ideology variable into the STIRPAT model is a logical extension. This approach allows for a more comprehensive analysis of the factors affecting e-waste recycling, considering not only traditional determinants like population, affluence, and technology but also the ideological stance of the government. By incorporating government ideology into the model, the study seeks to enhance our understanding of the nuanced factors shaping e-waste recycling rates in the European context. Accordingly, the following empirical models were constructed and estimated.

$$
\begin{aligned}
LOGEWASTE_{it} = {} & \beta_0 + \beta_1 LOGRPCGDP_{it} + \beta_2 LOGRPCGDP\char94 2_{it} + \beta_3 LOGPOPUL_{it} + \beta_4 LOGENINT_{it} \\
& + \beta_5 LOGEWCOLL_{it} + \beta_6 LOGGOVRIGHT_{it} + u_{it}
\end{aligned}
\tag{1}
$$

$$
\begin{aligned}
LOGEWASTE_{it} = {} & \beta_0 + \beta_1 LOGRPCGDP_{it} + \beta_2 LOGRPCGDP\char94 2_{it} + \beta_3 LOGPOPUL_{it} + \beta_4 LOGENINT_{it} \\
& + \beta_5 LOGEWCOLL_{it} + \beta_6 LOGGOVLEFT_{it} + u_{it}
\end{aligned}
\tag{2}
$$

$$
\begin{aligned}
LOGEWASTE_{it} = {} & \beta_0 + \beta_1 LOGRPCGDP_{it} + \beta_2 LOGRPCGDP\char94 2_{it} + \beta_3 LOGPOPUL_{it} + \beta_4 LOGENINT_{it} \\
& + \beta_5 LOGEWCOLL_{it} + \beta_6 LOGGOVCENTER_{it} + u_{it}
\end{aligned}
\tag{3}
$$

In Eqs 1, 2 and 3 above, $i$ and $t$ subscripts represent country and time, respectively. The reason for separately adding LOGGOVRIGHT, LOGGOVLEFT, and LOGGOVCENTER variables to the models is to avoid a potential multicollinearity problem. One-to-one summation of the observation values of LOGGOVRIGHT, LOGGOVLEFT, and LOGGOVCENTER variables is equal to 100 in most years across countries, which could lead to a multicollinearity problem in the model. We performed a Farrar-Glauber multicollinearity test and calculated the condition index, which yielded a test score of 2607.4241 (P-value: 0.0000) and a condition index value of 1413.27, respectively. Since the Farrar-Glauber multicollinearity test statistic is statistically significant, and the condition index value is far beyond the critical value of 30, we concluded that a multicollinearity problem exists. Therefore, it was more reasonable to include the variables LOGGOVRIGHT, LOGGOVLEFT, and LOGGOVCENTER separately. Logarithmic forms of all variables were utilized in the study.

The dependent variable in this study was the recycling rate of e-waste (EWASTE), measured as the percentage of e-waste that is recycled/reused out of the total waste electrical and electronic equipment collected in a country. The data for EWASTE was acquired from Eurostat, the statistical office of the European Union.

In the light of the IPAT and STIRPAT models, we employed total population (POPUL), real per capita gross domestic product (at constant 2010 Euro) (RPCGDP), and the energy intensity of the economy (kilograms of oil equivalent per thousand Euros) (ENINT) as control

variables. RPCGDP and ENINT data were obtained from Eurostat, while POPUL data was taken from the World Development Indicators (WDI) provided by the World Bank. According to the STIRPAT model, we expected the signs of these variables to be negative. That is, an increase in total population, GDP per capita, and energy intensity was anticipated to result in a decrease in the e-waste recycling rate.

The inclusion of the GDP per capita square (RPC^2) variable in the model aimed to test whether a non-linear relationship exists between economic development and the e-waste recycling rate, as suggested by the Environmental Kuznets Curve (EKC) hypothesis. According to this hypothesis, the relationship between economic development and environmental degradation could follow an inverted U-shaped curve, meaning that environmental degradation increases with economic development but eventually decreases after reaching a turning threshold level. The Environmental Kuznets Curve (EKC) hypothesis has been applied to various environmental issues to explore the relationship between economic development and environmental quality. Empirical studies examining the EKC Hypothesis have investigated various environmental indicators, including air pollution [29, 30], water pollution [31, 32], deforestation [33, 34], ecological footprint [35], and waste generation [36–41] across different countries or regions. In the case of the e-waste recycling rate, this relationship could be either linear or U-shaped, as the rate of e-waste recycling represents an environmental improvement. The U-shaped pattern will be confirmed if the coefficient for the LOGRPC variable is negative and the coefficient for LOGRPC^2 is positive and statistically significant.

In addition to the variables suggested by the IPAT and STIRPAT models, we also included total e-waste collected (kilograms per capita) (EWCOLL) as a control variable based on relevant literature [8, 42]. EWCOLL data was obtained from Eurostat. In our model, we expected a positive coefficient on this variable as it indicates more e-waste collected and suggests further recycling efforts to protect the environment.

Government ideology is the main variable of interest in this study. The literature on the impact of government ideology typically follows the approach of Bjørnskov [43, 44], which assigns a value of -1 to right-wing, 0 to centrist, and 1 to left-wing ideologies and weight single-party ideologies with their proportion of seats in the parliament. In this study, we employed real data of governing parties instead of assigning values of -1, 0, and 1 as in dummy variables. Our data of government ideology are given by the relative power position of relevant party wing (i.e., left-wing, right-wing, and center) in government based on their seat share in parliament, measured in percentage of the total parliamentary seat share of all governing parties. Hence, government ideology is represented by GOVRIGHT (i.e., relative power position of right-wing parties in government based on their seat share in parliament, measured in percentage of the total parliamentary seat share of all governing parties), GOVLEFT (i.e., relative power position of social democratic and other left parties in government based on their seat share in parliament, measured in percentage of the total parliamentary seat share of all governing parties), and GOVCENTER (i.e., relative power position of center parties in government based on their seat share in parliament, measured in percentage of the total parliamentary seat share of all governing parties) variables. We also affirmed the robustness of our findings by using two other indicators of government ideology (i.e., the percentage of cabinet posts held by the relevant party wing out of the total cabinet posts and the parliamentary seat share of the relevant party in the government). All government ideology data were collected from Armingeon et al. [45].

The data used in the analysis consist of unbalanced data from 30 European countries (28 EU member countries, Norway, and Iceland) and cover the period between 2008 and 2018. Table 1 provides the list of the variables used in the analysis and their descriptions and data sources. The limitations in the study's period and number of countries analyzed were

**Table 1. Definition and sources of variables.**

| Variable | Description | Source |
|---|---|---|
| ERATE | the e-waste recycling rate and given by the percentage of e-waste recycled/reused in a country. | The Statistical Office of the European Union (Eurostat) |
| RPC | real GDP per capita and measured by per capita gross domestic product (at constant 2010 Euro). | The Statistical Office of the European Union (Eurostat) |
| EINT | the energy intensity of the economy and expressed in kilograms of oil equivalent per thousand Euros. | The Statistical Office of the European Union (Eurostat) |
| ECOLL | e-waste collected in kilograms per inhabitant. | The Statistical Office of the European Union (Eurostat) |
| POP | total population. | World Development Indicators (WDI) |
| RIGHT | Relative power position of right-wing parties in government based on their seat share in parliament, measured in percentage of the total parliamentary seat share of all governing parties. | Armingeon et al. [45] |
| LEFT | Relative power position of left-wing parties in government based on their seat share in parliament, measured in percentage of the total parliamentary seat share of all governing parties. | Armingeon et al. [45] |
| CENTER | Relative power position of center parties in government based on their seat share in parliament, measured in percentage of the total parliamentary seat share of all governing parties. | Armingeon et al. [45] |

determined by the availability of data on e-waste recycling rates, which were only available for the period of 2008–2018 and for the 30 European countries with the largest sample size.

## Estimation technique

In this study, we utilized panel quantile regression method instead of conventional methods such as pooled OLS, Fixed Effect Model (FEM) regression, and Random Effect Model (REM) regression to estimate models presented in Eqs 1, 2 and 3. The use of quantile regression was motivated by several reasons. Firstly, the statistical distribution of data often exhibits unequal variation and thus the association between the relevant variables can differ depending on the locations of the dependent variable's conditional distribution. Therefore, using estimation methods relying on the mean values, such as pooled OLS, FEM, and REM, may yield incorrect results, as pointed out by Cade and Noon [46]. In other words, the panel quantile regression method relaxes the common regression slope assumption of the conventional OLS method and allows the slope to differ across quantiles. On the other hand, unlike the conventional OLS method, which presumes that relationships between dependent and independent variables are the same at all levels of data, the quantile regression method analyzes the different locations of the conditional distribution of the dependent variable (i.e., recycling rate of e-waste in our case) and, hence, obtains more robust results and a more comprehensive picture with detailed insights into the association between the recycling rate of e-waste and relevant independent variables. Put differently, the panel quantile regression method obtains more flexibility than the conventional OLS method to identify differing associations between independent and dependent variables at different parts of the distribution of the dependent variable. Additionally, as discussed in Hübler [47] and Chen and Lei [48], conventional OLS estimation method leads to biased estimators in the presence of outliers where one outlier may drastically alter estimation findings of OLS method while quantile regression method, where the conditional distribution of the dependent variable is divided into different quantiles, is immune to such bias and generates results more robust to outliers. Moreover, as stated in Xu and Lin [49], when the distribution of data is not normal distribution then estimators obtained from OLS

method are not consistent whereas estimators gathered from quantile regression method are consistent estimators irrespective of distributional assumption. Furthermore, when OLS method does not satisfy its assumptions (i.e., assumption of linearity, homoscedasticity, independence, and normality) then quantile regression method is an alternative to OLS method. Finally, the conventional OLS method employs the method of least squares to estimate the conditional mean of the dependent variable while quantile regression method utilizes the method of minimizing median absolute deviation to impute the conditional median or other quantiles of the dependent variable.

Albulescu et al. [50] have discussed that panel quantile regression with fixed effects is not an appropriate and efficient estimation method when T (11 years in our sample) is small and N (30 countries in our sample) is large. Our sample data comply with the warning of Albulescu et al. [50]. Therefore, we employed panel quantile regression with non-additive fixed effects introduced by Powell [51] to overcome this problem. Similar to the studies conducted by Xu et al. [52] and Boubellouta and Kusch-Brandt [8], we estimated and reported the findings for five quantiles, namely 10th, 25th, 50th, 75th, and 90th quantiles of the conditional distribution of e-waste recycling rate. In this case, the lower quantiles were represented by the countries with lower e-waste recycling rates, while the upper quantiles were represented by the countries with higher e-waste recycling rates. We utilized the adaptive Markov Chain Monte Carlo (MCMC) optimization method in our estimations.

## Estimation results

Table 2 presents the estimation results of the government ideology model for right-wing parties (i.e., LOGGOVRIGHT variable) as given by Eq 1.

We identified a statistically significant negative relationship between the variables LOGGOVRIGHT and LOGEWASTE for all five quantiles of the distribution of e-waste recycling rate. In other words, regardless of the quantile we assess, an increase in the relative power position of right-wing parties in government was associated with a decrease in e-waste recycling

**Table 2. Quantile regression results for Eq 1.**

|  | 10th quantile | 25th quantile | 50th quantile | 75th quantile | 90th quantile |
|---|---|---|---|---|---|
| LOGRPCGDP | -5.5467 | -1.8114 | -4.8836 | -2.1651 | -1.8341 |
| *P-value* | *0.0000* | *0.0000* | *0.0000* | *0.0000* | *0.0000* |
| LOGRPCGDP^2 | 0.2373 | 0.0528 | 0.2090 | 0.0776 | 0.0566 |
| *P-value* | *0.0000* | *0.0000* | *0.0000* | *0.0000* | *0.0000* |
| LOGENINT | -0.2809 | -0.3042 | -0.2728 | -0.2731 | -0.3318 |
| *P-value* | *0.0000* | *0.0000* | *0.0000* | *0.0000* | *0.0000* |
| LOGPOPUL | -0.0276 | -0.0382 | -0.0786 | -0.0774 | -0.0796 |
| *P-value* | *0.0000* | *0.0000* | *0.0000* | *0.0000* | *0.0000* |
| LOGEWCOLL | 1.1586 | 1.0254 | 0.9180 | 0.8692 | 0.9370 |
| *P-value* | *0.0000* | *0.0000* | *0.0000* | *0.0000* | *0.0000* |
| LOGGOVRIGHT | -0.0041 | -0.0013 | -0.0045 | -0.0005 | -0.0028 |
| *P-value* | *0.0000* | *0.0000* | *0.0000* | *0.0770* | *0.0000* |
| Wald test | 40387.62 | 1.10E+06 | 79457.06 | 2.40E+06 | 7.20E+05 |
| *P-value* | *0.0000* | *0.0000* | *0.0000* | *0.0000* | *0.0000* |
| Number of obs. | 290 | 290 | 290 | 290 | 290 |
| Number of countries | 30 | 30 | 30 | 30 | 30 |
| Min. obs. per country | 5 | 5 | 5 | 5 | 5 |
| Max. obs. per country | 11 | 11 | 11 | 11 | 11 |

rates. The estimation results provide support for the hypothesis of the partisan theory (research hypothesis 2), which suggests that discernible differences exist among governments in their approaches to the rate of e-waste recycling. The findings substantiate the argument put forth by the partisan theory, indicating that right-wing parties place less emphasis on environmental concerns, environmental quality, and the protection of the environment compared to their left-wing counterparts.

On the other hand, we did not identify a systematic pattern for the association between LOGGOVRIGHT and LOGEWASTE variables across quantiles. The magnitude of the significant negative impact of right-wing governments on e-waste recycling rates in absolute terms decreased between the 10th and 25th quantiles, increased between the 25th and 50th quantiles, diminished between the 50th and 75th quantiles, and further increased between the 75th and 90th quantiles. Meanwhile, we have observed that the largest negative effect of right-wing governments on e-waste recycling rates occurred at the median (i.e., 50th quantile). This reveals that right-wing governments had a more pronounced negative impact on the e-waste recycling rate in economies where the e-waste recycling rate (i.e., e-waste recycling management level) was at a medium level compared to low-level and high-level recycling management.

The estimation results in Table 2 indicate statistically significant negative and positive coefficients for the per capita real GDP and squared per capita real GDP variables, respectively, across all five quantiles. This implies that there exists a U-shaped relationship between the e-waste recycling rate, which is a measure of environmental improvement, and the per capita real GDP, which reflects the level of development in a country. This significant U-shaped relationship between the e-waste recycling rate and per capita real GDP is consistent with and confirms the well-established inverted U-shaped Environmental Kuznets Curve (EKC) hypothesis, which suggests a positive relationship between development and environmental degradation at low-income levels and a negative relationship between development and environmental degradation at high-income levels. Our study demonstrates that there exists a U-shaped relationship between the e-waste recycling rate and real GDP per capita. This finding is consistent with previous research on solid waste, which has identified an inverted U-shaped relationship between GDP per capita and solid waste generation (i.e., environmental degradation) (see, for example, [40, 41] for e-waste). Additionally, our findings support Cerqueira and Soukiazis's [42] research on recycling levels in Portuguese municipalities, which also demonstrated a U-shaped relationship between gross-value added per capita and recycling level per capita.

As observed in Table 2, population had a statistically significant negative impact on e-waste recycling rate across all quantiles of the distribution of e-waste recycling rate. This implies that as population increases, e-waste recycling rates tended to decrease. One possible explanation for this trend could be the insufficiency of e-waste recycling capacity to effectively cope with the rising population, resulting in lower recycling rates, as discussed by Adshead et al. [53], Churchill et al. [54], and Hummel and Lux [55]. This highlights the need for adequate infrastructure and capacity building in e-waste management to accommodate the growing population and promote sustainable recycling practices.

The energy intensity variable consistently exhibited a statistically significant negative coefficient across all quantiles. This finding indicates that e-waste recycling rate decreased in response to increases in energy intensity. This finding aligns with the perspective proposed by Morley et al. [56], which suggests that as the usage of electrical and electronic equipment increases, energy intensity also tends to rise. Consequently, this may result in higher generation of e-wastes, which could potentially remain unprocessed due to limitations in the existing e-waste management system of the relevant country.

A statistically significant positive coefficient was observed for the collected e-waste variable in all five quantiles, indicating that the recycling rate of e-waste tended to increase as the

amount of collected e-waste rises. This finding is consistent with the studies conducted by Boubellouta and Kusch-Brandt [41] and Cerqueira and Soukiazis [42], which showed that the amount of uncollected e-waste was a significant factor influencing the amount of non-recycled e-waste in European countries and Spanish municipalities, respectively.

Table 2 also presents the results of the Wald test (Koenker and Bassett [57]), which tested the statistical significance of the model and assumed the null hypothesis that all the partial slope coefficients of the model were equal to zero (i.e., the model was statistically non-significant). The results of the Wald tests indicated that all five estimated models were statistically significant.

The estimation results of the government ideology model for left-wing parties (i.e., LOG-GOVLEFT variable) as given by Eq 2 are shown in Table 3.

Table 3 indicated a statistically significant positive relationship between the variables LOG-GOVLEFT and LOGEWASTE for all five quantiles of the distribution of the e-waste recycling rate. In other words, an increase in the relative power position of left-wing parties in government was associated with an augmentation in e-waste recycling rates across all quantiles. The results of the estimation robustly back the research hypothesis posited by the partisan theory (Hypothesis 2), affirming the notion that noticeable distinctions prevail among governments in their strategies for the rate of e-waste recycling. These findings strongly substantiate the overarching argument advanced by the partisan theory, revealing that left-wing parties prioritize environmental concerns, environmental quality, and environmental protection to a greater extent than right-wing parties.

However, there was no a systematic pattern for the relationship between LOGGOVLEFT and LOGEWASTE variables across quantiles. The magnitude of the significant positive impact of left-wing governments on e-waste recycling rates decreased between the 10th and 25th quantiles, increased between the 25th and 50th quantiles, and further increased between the 50th and 75th quantiles, but declined between the 75th and 90th quantiles. In addition, we observed that the largest positive effect of left-wing governments on e-waste recycling rates

**Table 3. Quantile regression results for Eq 2.**

|  | 10th quantile | 25th quantile | 50th quantile | 75th quantile | 90th quantile |
|---|---|---|---|---|---|
| LOGRPCGDP | -3.1559 | -1.7453 | -1.3213 | -1.8292 | -1.5905 |
| *P-value* | *0.0000* | *0.0000* | *0.0000* | *0.0000* | *0.0000* |
| LOGRPCGDP^2 | 0.1054 | 0.0492 | 0.0219 | 0.0631 | 0.0491 |
| *P-value* | *0.0000* | *0.0000* | *0.0000* | *0.0000* | *0.0000* |
| LOGENINT | -0.6279 | -0.2993 | -0.5628 | -0.1823 | -0.3201 |
| *P-value* | *0.0000* | *0.0000* | *0.0000* | *0.0000* | *0.0000* |
| LOGPOPUL | -0.0354 | -0.0292 | -0.1443 | -0.0938 | -0.0795 |
| *P-value* | *0.0000* | *0.0000* | *0.0000* | *0.0000* | *0.0000* |
| LOGEWCOLL | 1.1082 | 1.0296 | 0.9516 | 0.9097 | 0.8534 |
| *P-value* | *0.0000* | *0.0000* | *0.0000* | *0.0000* | *0.0000* |
| LOGGOVLEFT | 0.0148 | 0.0021 | 0.0023 | 0.0038 | 0.0007 |
| *P-value* | *0.0000* | *0.0000* | *0.0000* | *0.0000* | *0.0000* |
| Wald test | 30920.54 | 1.90E+05 | 69915.16 | 3.10E+05 | 2.50E+07 |
| *P-value* | *0.0000* | *0.0000* | *0.0000* | *0.0000* | *0.0000* |
| Number of obs. | 290 | 290 | 290 | 290 | 290 |
| Number of countries | 30 | 30 | 30 | 30 | 30 |
| Min. obs. per country | 5 | 5 | 5 | 5 | 5 |
| Max. obs. per country | 11 | 11 | 11 | 11 | 11 |

occurred at the 10th quantile, while the smallest positive effect occurred at the 90th quantile. This indicated that left-wing governments had the highest positive impact on the e-waste recycling rate in economies where the e-waste recycling management level (i.e., e-waste recycling rate) was at the lowest level (i.e., 10th quantile), while left-wing governments had the lowest positive impact on the e-waste recycling rate in economies where the e-waste recycling management level (i.e., e-waste recycling rate) was at the highest level (i.e., 90th quantile).

In relation to the control variables, the results presented in Table 3 affirmed the findings reported in Table 2. In Table 3, we observed a statistically significant U-shaped association between e-waste recycling rate and per capita real GDP across all five quantiles. This finding supports the validity of the U-shaped EKC hypothesis, as evidenced by the negative coefficients of log per capita real GDP and positive coefficients of log squared per capita real GDP. Regarding the remaining independent variables of the model, our coefficient estimations were in line with our prior expectations across all quantiles. The coefficients of the population and energy intensity variables exhibited statistically significant negative signs, while the coefficient of the collected e-waste variable displayed a statistically significant positive sign in all quantiles. Additionally, Wald test results indicated that all five estimated models were statistically significant.

Table 4 presented the estimation results of the government ideology model for central parties (i.e., LOGGOVCENTER variable) as given by Eq 3.

As evident from Table 4, there was a statistically significant positive correlation between the LOGGOVCENTER variable and the LOGEWASTE variable for all five quantiles of the distribution of e-waste recycling rate. More specifically, we observed that increases in the relative power position of central parties in government led to an expansion of e-waste recycling rates in all five estimated models. The estimation results once again strongly support the research hypothesis proposed by the partisan theory (Hypothesis 2), confirming the idea that significant differences exist among governments in their approaches to the rate of e-waste recycling. The traditional understanding of the Partisan Theory relies on the left-right dichotomy as a fundamental analytical framework, categorizing governing political parties strictly as either left or

**Table 4. Quantile regression results for Eq 3.**

|  | 10th quantile | 25th quantile | 50th quantile | 75th quantile | 90th quantile |
|---|---|---|---|---|---|
| LOGRPCGDP | -1.6382 | -1.7481 | -1.8852 | -1.9339 | -1.8482 |
| *P-value* | *0.0000* | *0.0000* | *0.0000* | *0.0000* | *0.0000* |
| LOGRPCGDP^2 | 0.0391 | 0.0457 | 0.0594 | 0.0649 | 0.0618 |
| *P-value* | *0.0000* | *0.0000* | *0.0000* | *0.0000* | *0.0000* |
| LOGENINT | -0.4822 | -0.3487 | -0.2997 | -0.2757 | -0.2820 |
| *P-value* | *0.0000* | *0.0000* | *0.0000* | *0.0000* | *0.0000* |
| LOGPOPUL | -0.0160 | -0.0135 | -0.0540 | -0.0859 | -0.0909 |
| *P-value* | *0.0000* | *0.0000* | *0.0000* | *0.0000* | *0.0000* |
| LOGEWCOLL | 0.9845 | 1.0522 | 1.0048 | 0.8432 | 0.8730 |
| *P-value* | *0.0000* | *0.0000* | *0.0000* | *0.0000* | *0.0000* |
| LOGGOVCENTER | 0.0223 | 0.0161 | 0.0114 | 0.0086 | 0.0071 |
| *P-value* | *0.0000* | *0.0000* | *0.0000* | *0.0000* | *0.0000* |
| Wald test | 8.30E+06 | 1.50E+05 | 75457.32 | 4.90E+06 | 3.50E+06 |
| *P-value* | *0.0000* | *0.0000* | *0.0000* | *0.0000* | *0.0000* |
| Number of obs. | 290 | 290 | 290 | 290 | 290 |
| Number of countries | 30 | 30 | 30 | 30 | 30 |
| Min. obs. per country | 5 | 5 | 5 | 5 | 5 |
| Max. obs. per country | 11 | 11 | 11 | 11 | 11 |

right. Nevertheless, our analysis reveals the noteworthy influence of center-wing parties in advancing e-waste recycling rates. This contributes valuable additional insights into the role of center-wing parties in this context.

Additionally, we identified a systematic pattern in the relationship between the LOGGOV-CENTER and LOGEWASTE variables across the quantiles, where the positive impact of central governments diminished for each consecutive quantile. The largest positive effect of central governments on e-waste recycling rate was observed in the 10th quantile, indicating that central governments had the highest positive impact on e-waste recycling rate in economies where the e-waste recycling management level (i.e., e-waste recycling rate) was at its lowest (i.e., 10th quantile).

As depicted in Table 4, mirroring our earlier estimation results in Table 2, we identified a statistically significant U-shaped relationship between log e-waste recycling rate and log per capita real GDP, confirming the validity of the U-shaped EKC hypothesis across all quantiles. Similarly, we found a statistically significant negative coefficient for the population and energy intensity variables, and a positive coefficient for the collected e-waste variable, aligning with our prior expectations. Furthermore, the Wald test confirms the significance of all five estimated models.

## Discussion

Based on our estimations, we found that an increase in the relative power of right-wing parties within the government was associated with a decrease in the rate of e-waste recycling. Conversely, an increase in the relative power of left-wing or center-wing parties was associated with an increase in the rate of e-waste recycling. This relationship held consistently across different quantiles analyzed. These findings align with the expectations of the partisan theory, which suggests that ideological disparities among political parties can result in divergent policy preferences and outcomes.

This study contributes a pioneering analysis in the literature by being the first to explore the influence of government ideology on e-waste recycling rates. Previous research (Boubellouta and Kusch-Brand [8], Yilmaz and Koyuncu [9], Constantinescu et al. [10]) has overlooked government ideology as a factor impacting e-waste recycling, and this study addresses this gap by introducing a new determinant affecting e-waste recycling rates. As far as our knowledge extends, no empirical research has investigated the relationship between government ideology and e-waste recycling rates. Thus, this study significantly contributes to the literature by identifying government ideology as a potential determinant of the rate of e-waste recycling, filling a substantial research gap in this domain.

This study also contributes to the current research on the relationship between government ideology and the environmental quality. Earlier research has overlooked the significance of e-waste recycling rates as an indicator of environmental quality. This study extends the current body of research examining the interplay between government ideology and the environment. It introduces new evidence that underscores the influence of government ideology on environmental progress, specifically in the context of e-waste recycling rates. Our estimations, using real measures of governing parties, largely align with previous cross-country studies of King and Borchardt [11], Bernauer and Koubi [13], Chang et al. [15], Wen et al. [17], and Cetkovic and Hagemann [19]. Using the conventional left-right dichotomy, where right-wing and center parties are categorized under the right and measuring environmental quality by proxies such as greenhouse gas emissions, overall environmental quality, the Environmental Performance Index, and climate policy, these studies demonstrated that left-wing parties prioritize environmental quality more than right-wing parties.

Our analysis also sheds light on the significant impact of center-wing parties on environmental quality, particularly in promoting e-waste recycling rates. This provides valuable additional insights into the role of center-wing parties within this context. Our estimation results align with the findings of Garmann [14] and Kammerlander and Schulzetest [18], who provided evidence regarding the role of center-wing governments in reducing carbon dioxide emissions and enhancing the Environmental Performance Index (EPI), respectively.

In order to the check of robustness of our findings, we used two other indicators of government ideology, namely the percentage of cabinet posts held by the relevant party wing out of the total cabinet posts and the parliamentary seat share of the relevant party in the government.

The data for government ideology indicators were taken from Armingeon et al. [45]. The estimation results of the government ideology model for right-wing parties (LOGGOVRIGHT variable), left-wing parties (LOGGOVLEFT variable), and central parties (LOGGOVCENTER variable) regarding two other indicators of government ideology were presented in Tables 5–7, respectively.

By utilizing the same sample size (i.e., 290 observations from 30 countries) and incorporating two additional indicators of government ideology, we identified a statistically significant negative relationship between the LOGGOVRIGHT and LOGEWASTE variables and a statistically significant positive relationship between the LOGGOVLEFT, LOGGOVCENTER, and LOGEWASTE variables for selected quantiles of the distribution of e-waste recycling rate, as observed in Tables 5–7. Thus, our estimation results using two other indicators of government ideology supported our previous findings and indicated the robustness of our findings. Therefore, our estimation results, employing two other indicators of government ideology, supported our earlier findings and underscored the robustness of our results.

To summarize, our study emphasizes the substantial impact of political ideology on e-waste recycling rates, highlighting the imperative for continued research in this domain. This

**Table 5. Quantile regression results for Eq 1 for two other indicators of government ideology.**

| | Indicator: The percentage of cabinet posts held by the relevant party wing out of the total cabinet posts. | | | Indicator: The parliamentary seat share of the relevant party in the government. | | |
|---|---|---|---|---|---|---|
| | 10th quantile | 50th quantile | 90th quantile | 10th quantile | 50th quantile | 90th quantile |
| LOGRPCGDP | -10,5073 | -4,3113 | -1,5672 | -3,7150 | -2,2839 | -1,2784 |
| *P-value* | *0,0000* | *0,0020* | *0,0000* | *0,0000* | *0,0000* | *0,0000* |
| LOGRPCGDP^2 | 0,4606 | 0,1810 | 0,0412 | 0,1503 | 0,0768 | 0,0337 |
| *P-value* | *0,0000* | *0,0080* | *0,0000* | *0,0000* | *0,0000* | *0,0000* |
| LOGENINT | -0,7272 | -0,2834 | -0,3387 | -0,2001 | -0,2989 | -0,2778 |
| *P-value* | *0,0000* | *0,0000* | *0,0000* | *0,0000* | *0,0000* | *0,0000* |
| LOGPOPUL | -0,1944 | -0,0866 | -0,0943 | -0,0334 | -0,0879 | -0,0574 |
| *P-value* | *0,0000* | *0,0000* | *0,0000* | *0,0000* | *0,0000* | *0,0000* |
| LOGEWCOLL | 1,5294 | 0,9282 | 0,9219 | 0,9683 | 0,9755 | 0,8612 |
| *P-value* | *0,0000* | *0,0000* | *0,0000* | *0,0000* | *0,0000* | *0,0000* |
| LOGGOVRIGHT | -0,0101 | -0,0033 | -0,0057 | -0,0051 | -0,0078 | -0,0034 |
| *P-value* | *0,0090* | *0,0010* | *0,0000* | *0,0520* | *0,0000* | *0,0000* |
| Wald test | 1,95E+04 | 2,99E+04 | 2,20E+05 | 1,59E+04 | 7,01E+04 | 7,02E+04 |
| *P-value* | *0,0000* | *0,0000* | *0,0000* | *0,0000* | *0,0000* | *0,0000* |
| Number of obs. | 290 | 290 | 290 | 290 | 290 | 290 |
| Number of countries | 30 | 30 | 30 | 30 | 30 | 30 |
| Min. obs. per country | 5 | 5 | 5 | 5 | 5 | 5 |
| Max. obs. per country | 11 | 11 | 11 | 11 | 11 | 11 |

**Table 6. Quantile regression results for Eq 2 for two other indicators of government ideology.**

| | Indicator: The percentage of cabinet posts held by the relevant party wing out of the total cabinet posts. | | | Indicator: The parliamentary seat share of the relevant party in the government. | | |
|---|---|---|---|---|---|---|
| | **10th quantile** | **50th quantile** | **90th quantile** | **10th quantile** | **50th quantile** | **90th quantile** |
| LOGRPCGDP | -2,1525 | -2,0185 | -1,7506 | -5,0255 | -2,0892 | -1,8929 |
| *P-value* | *0,0000* | *0,0000* | *0,0000* | *0,0000* | *0,0000* | *0,0000* |
| LOGRPCGDP^2 | 0,0633 | 0,0700 | 0,0574 | 0,2050 | 0,0724 | 0,0664 |
| *P-value* | *0,0000* | *0,0000* | *0,0000* | *0,0000* | *0,0000* | *0,0000* |
| LOGENINT | -0,4085 | -0,3842 | -0,3077 | -0,7006 | -0,2391 | -0,3174 |
| *P-value* | *0,0000* | *0,0000* | *0,0000* | *0,0000* | *0,0000* | *0,0000* |
| LOGPOPUL | -0,0160 | -0,0980 | -0,0785 | -0,0282 | -0,0657 | -0,0724 |
| *P-value* | *0,0000* | *0,0000* | *0,0000* | *0,0000* | *0,0000* | *0,0000* |
| LOGEWCOLL | 1,1108 | 0,8317 | 0,8605 | 1,2821 | 0,9103 | 0,8489 |
| *P-value* | *0,0000* | *0,0000* | *0,0000* | *0,0000* | *0,0000* | *0,0000* |
| LOGGOVLEFT | 0,0173 | 0,0014 | 0,0002 | 0,0032 | 0,0015 | 0,0006 |
| *P-value* | *0,0000* | *0,0660* | *0,0010* | *0,0770* | *0,0000* | *0,0000* |
| Wald test | 1,80E+07 | 4,63E+04 | 4,40E+07 | 1,20E+06 | 1,10E+05 | 4,20E+07 |
| *P-value* | *0,0000* | *0,0000* | *0,0000* | *0,0000* | *0,0000* | *0,0000* |
| Number of obs. | 290 | 290 | 290 | 290 | 290 | 290 |
| Number of countries | 30 | 30 | 30 | 30 | 30 | 30 |
| Min. obs. per country | 5 | 5 | 5 | 5 | 5 | 5 |
| Max. obs. per country | 11 | 11 | 11 | 11 | 11 | 11 |

**Table 7. Quantile regression results for Eq 3 for two other indicators of government ideology.**

| | Indicator: The percentage of cabinet posts held by the relevant party wing out of the total cabinet posts. | | | Indicator: The parliamentary seat share of the relevant party in the government. | | |
|---|---|---|---|---|---|---|
| | **10th quantile** | **50th quantile** | **90th quantile** | **10th quantile** | **50th quantile** | **90th quantile** |
| LOGRPCGDP | -1,7165 | -2,6032 | -1,6078 | -1,9175 | -2,4633 | -5,2428 |
| *P-value* | *0,0000* | *0,0000* | *0,0000* | *0,0000* | *0,0000* | *0,0000* |
| LOGRPCGDP^2 | 0,0428 | 0,0953 | 0,0497 | 0,0565 | 0,0881 | 0,2295 |
| *P-value* | *0,0000* | *0,0000* | *0,0000* | *0,0000* | *0,0000* | *0,0000* |
| LOGENINT | -0,6271 | -0,3072 | -0,3190 | -0,4909 | -0,3098 | -0,1660 |
| *P-value* | *0,0000* | *0,0000* | *0,0000* | *0,0000* | *0,0000* | *0,0000* |
| LOGPOPUL | -0,0249 | -0,0777 | -0,0728 | -0,0266 | -0,0809 | -0,0419 |
| *P-value* | *0,0000* | *0,0000* | *0,0000* | *0,0000* | *0,0000* | *0,0000* |
| LOGEWCOLL | 1,0390 | 0,9097 | 0,8566 | 0,9247 | 0,9126 | 0,9268 |
| *P-value* | *0,0000* | *0,0000* | *0,0000* | *0,0000* | *0,0000* | *0,0000* |
| LOGGOVCENTER | 0,0142 | 0,0139 | 0,0021 | 0,0198 | 0,0168 | 0,0091 |
| *P-value* | *0,0000* | *0,0000* | *0,0010* | *0,0000* | *0,0000* | *0,0000* |
| Wald test | 8,70E+05 | 1,37E+04 | 4,30E+05 | 1,30E+07 | 1,04E+04 | 4,15E+04 |
| *P-value* | *0,0000* | *0,0000* | *0,0000* | *0,0000* | *0,0000* | *0,0000* |
| Number of obs. | 290 | 290 | 290 | 290 | 290 | 290 |
| Number of countries | 30 | 30 | 30 | 30 | 30 | 30 |
| Min. obs. per country | 5 | 5 | 5 | 5 | 5 | 5 |
| Max. obs. per country | 11 | 11 | 11 | 11 | 11 | 11 |

emphasis on further exploration will not only contribute to a deeper understanding of the intricate dynamics between political ideology and recycling practices but also provide essential insights to inform and shape well-informed policy-making in the realm of e-waste management.

## Conclusions

This study aimed to examine the impact of government ideology on e-waste recycling rates across 30 European countries, including 28 EU member countries, Norway, and Iceland. This examination involved scrutinizing the validity of both the consensus hypothesis and the partisan theory. The consensus hypothesis posits that governments exhibit similar approaches to e-waste recycling rates, while the partisan theory argues that government ideology matters, with left-wing governments being more inclined to promote the rate of e-waste recycling compared to their right-wing counterparts. To conduct this research, we employed an unbalanced panel dataset spanning from 2008 to 2018 and utilized the panel quantile regression method.

The results of our study confirmed the expectations put forth by the partisan theory in relation to e-waste recycling rates. Specifically, our findings indicated that when right-wing parties gained more relative power in government, there was a decrease in the rate of e-waste recycling. On the other hand, an increase in the relative power of left-wing or center-wing parties in government was associated with an increase in e-waste recycling rates across all examined quantiles. These results align with the predictions of the partisan theory, which suggests that ideological variations among political parties can lead to divergent policy preferences and outcomes. Furthermore, the estimation results remained robust when different indicators of government ideology were considered.

Our study's findings have significant implications for policymakers, emphasizing the vital role of political factors in influencing the rate of e-waste recycling. Although the European Union (EU) has made commendable progress in enhancing the rate of e-waste recycling, there remain persistent challenges that require further attention and resolution. One such challenge is the inadequate collection and proper disposal of electronic waste. While the EU has implemented regulations and directives to promote e-waste recycling, ensuring widespread compliance and effective enforcement remains a persistent issue. Insufficient infrastructure and resources for e-waste management, especially in certain regions or member states, further hinder the efficient recycling of electronic waste. Additionally, the lack of standardized procedures and harmonized practices across EU member states poses obstacles to seamless cross-border e-waste recycling, leading to inconsistencies and inefficiencies in the overall recycling process. The effective execution of policies aimed at promoting e-waste recycling rates in EU countries heavily relies on the political position of the governing body. Governments that demonstrate a strong dedication to advancing e-waste recycling rates are more inclined to develop and implement policies and effective strategies for extended producer responsibility (EPR), promoting eco-design principles, and implementing innovative recycling technologies in the EU.

The primary limitation of this study is the availability of data, and obtaining more comprehensive data on government ideology and political factors could enhance the study's results. Despite this limitation, our findings provide evidence of the significance of government ideology in promoting e-waste recycling rates in EU countries. These findings have implications for future research in this area and contribute to a deeper understanding of the connection between government ideology and sustainability. Subsequent studies could consider examining individual countries or regions, incorporating diverse political and institutional determinants of e-waste recycling rates as control variables, and utilizing various econometric techniques.

## Supporting information

**S1 File.**
(XLSX)

## Author Contributions

**Conceptualization:** Erdal Arslan, Cuneyt Koyuncu, Rasim Yilmaz.

**Formal analysis:** Erdal Arslan, Cuneyt Koyuncu, Rasim Yilmaz.

**Investigation:** Erdal Arslan, Rasim Yilmaz.

**Methodology:** Cuneyt Koyuncu.

**Writing – review & editing:** Erdal Arslan, Cuneyt Koyuncu, Rasim Yilmaz.

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
