## [Decision Letter · Decision Letter 0]

16 Oct 2023

PONE-D-23-29738The Influence of Government Ideology on the Rate of E-Waste Recycling in the European Union CountriesPLOS ONE

Dear Dr. Yilmaz,

Thank you for submitting your manuscript to PLOS ONE. After careful consideration, we feel that it has merit but does not fully meet PLOS ONE’s publication criteria as it currently stands. Therefore, we invite you to submit a revised version of the manuscript that addresses the points raised during the review process. Please see reviewers' comments below and address all of them carefully. Elaborate a response letter. Mark in color all you add into the revised text.

We look forward to receiving your revised manuscript.

Kind regards,

Magdalena Radulescu

Academic Editor

PLOS ONE

Journal Requirements:

Reviewers' comments:

Reviewer's Responses to Questions

**Comments to the Author**

1. Is the manuscript technically sound, and do the data support the conclusions?

Reviewer #1: Partly

Reviewer #2: Yes

2. Has the statistical analysis been performed appropriately and rigorously? 

Reviewer #1: No

Reviewer #2: Yes

3. Have the authors made all data underlying the findings in their manuscript fully available?

Reviewer #1: Yes

Reviewer #2: Yes

4. Is the manuscript presented in an intelligible fashion and written in standard English?

Reviewer #1: No

Reviewer #2: Yes

5. Review Comments to the Author

Reviewer #1: I am writing about the manuscript (PONE-D-23-29738) entitled “The Influence of Government Ideology on the Rate of E-Waste Recycling in the European Union Countries” Thank you for the opportunity to review this paper. The paper is not well-structured and conveys a deal of information. I recommend for the paper substantial modifications and refinements of the present version. My comments are as follows:

1. To enhance the quality of your abstract, you should add one or two sentences related to, why this research is important and how it may inform future policies or actions related to e-waste recycling in European countries.

2. The reader can understand your contribution, but the other explanation of the introduction part is not in the story flow. You need to describe it in more detail and justify your introduction.

3. Provide a clear rationale for the research design and the selection of the specific countries or cases in the introduction section or you can add it wherever it is suitable.

4. The EKC hypothesis is also analyzed by the author, but I couldn’t see any literature review on EKC.

5. Why do you use the IPAT and STIRPAT models for the selected variables? You should justify your study variables which are used in the IPAT and STIRPAT.

6. You mentioned in line 274 in the research methodology and data section that logarithmic forms of all variables were utilized in the study, but you didn’t write logarithmic form equations. You must write logarithmic form equations.

7. In the interpretation of Table 2 and Table 3 findings, you didn’t give any evidence to justify your findings. You need to improve your interpretation of Table 3 and justify your results of the mentioned tables.

8. Consider expanding the discussion section to provide a more in-depth analysis of the implications of your findings.

9. Include a roadmap from a selection of variables to the conclusion of the study so that the reader can understand easily.

10. There are many grammatical mistakes, you need to proofread to remove it.

Reviewer #2: I have reviewed your manuscript titled "The Influence of Government Ideology on the Rate of E-Waste Recycling in the European Union Countries" with great interest. The topic is relevant and timely, addressing the crucial issue of e-waste recycling in the context of government ideology. However, I have a few concerns and suggestions that need to be addressed for the manuscript to be considered for publication.

The manuscript utilizes the panel quantile regression method for analysis. While this method can be insightful, its choice over conventional methods such as pooled OLS needs a more comprehensive explanation. Please elaborate on why panel quantile regression was chosen and how it offers advantages over other methods in the context of your research question. Discuss the suitability of this method for handling the specific characteristics of your data.The manuscript lacks a detailed description of the data sources and variables used in the analysis. Transparency in data sources is vital for the reproducibility and reliability of your study.

The manuscript would benefit from a more explicit discussion of the theoretical framework guiding your research. Clearly state the hypotheses derived from the theoretical foundation. How does government ideology influence e-waste recycling rates according to existing literature, and what specific hypotheses are being tested in your study? A robust theoretical foundation will strengthen the conceptual framework of your research.For example, QBy simulating the behavior of various materials at the quantum level, researchers can identify processes that maximize material recovery while minimizing waste..uantum computing has the potential to revolutionize many areas, including environmental initiatives like e-waste recycling. While quantum computing is still in its infancy, researchers are exploring its applications in solving complex problems related to recycling, materials science, and environmental sustainability.

See the following, "Opportunities and constraints for developing a sustainable e-waste management system at local government level in Australia." Waste Management & Research 28, no. 8 (2010): 705-713

Provide a detailed interpretation of the results obtained from the panel quantile regression analysis. How do the findings align with your hypotheses? Discuss the practical implications of the results in the context of e-waste management policies. Additionally, consider comparing and contrasting your results with previous studies in the field to highlight the novelty and contributions of your work. The conclusion should succinctly summarize the key findings and their implications. Clearly outline the contributions of your study to the existing literature. Furthermore, discuss the limitations of your research and propose directions for future studies.

6. PLOS authors have the option to publish the peer review history of their article (what does this mean?). If published, this will include your full peer review and any attached files.

Reviewer #1: No

Reviewer #2: No

---

## [Author Response · Author response to Decision Letter 0]

18 Dec 2023

As per the suggestion of the esteemed referee/reviewer 1,

1) We have added sentences to the abstract that elaborate on "why this research is important and how it may inform future policies or actions related to e-waste recycling in European countries". Page 2, abstract, sentences are highlighted in yellow.

2) The introduction part of the study has been enhanced by providing a more detailed and justified explanation to better align with the story flow. Page 3-4, paragraphs are highlighted in yellow.

3) We have provided a clear rationale for the research design and the selection of specific countries in the introduction section. Page 3-4, paragraphs are highlighted in yellow.

4) We have provided a literature review on the Environmental Kuznets Curve (EKC) hypothesis. Page14, sentences are highlighted in yellow.

5) We have addressed the inquiry regarding the utilization of the IPAT and STIRPAT models for the selected variables by providing a justification for the inclusion of specific study variables within these frameworks. Page 12-13, the paragraph is highlighted in yellow.

6) The equations have been written in logarithmic forms. Page 13, equations are highlighted in blue.

7) We have improved the interpretations of Tables 2-4 by providing evidence and thorough justifications for the presented results. Page 19, 23-25, sentences are highlighted in yellow.

8) A more in-depth analysis elucidating the implications of the findings has been incorporated into the discussion section. Page 26-29, paragraphs are highlighted in yellow.

9) Proofreading has been done.

As per the suggestion of the esteemed referee/reviewer 2,

1) We have provided a more comprehensive explanation for choosing panel quantile regression over conventional methods, such as pooled OLS. The rationale behind selecting panel quantile regression and its advantages over other methods in the context of our research question are elaborated. Additionally, we discuss the suitability of this method for handling the specific characteristics of our data. Page 17 and 18, sentences are highlighted in yellow.

2) A detailed description of the data sources and variables used in the analysis have been provided. Additionally, a summary table has also been included. Pages 14-16, sentences are highlighted in green; Table 1 in page 16.

3) The hypotheses derived from the theoretical foundation have been clearly stated. “How does government ideology influence e-waste recycling rates according to existing literature, and what specific hypotheses are being tested in the study” have been explained. Pages 10-11, sentences are highlighted in yellow.

4) A detailed interpretation of the results obtained from the panel quantile regression analysis and a discussion on how the findings align with our hypotheses have been provided. Pages 19, 22, 24, sentences are highlighted in yellow.

5) We have compared our estimation results with previous studies in the literature to emphasize the novelty and contributions of our study. Pages 26-27, sentences are highlighted in yellow.

6) The key findings and their implications have been summarized and the contributions of our study to the existing literature have been outlined in the conclusion section. Page 30, sentences are highlighted in yellow.

7) The limitations of our research have been provided and directions for future studies have been proposed. Page 31, sentences are highlighted in yellow.

---

## [Decision Letter · Decision Letter 1]

4 Jan 2024

The Influence of Government Ideology on the Rate of E-Waste Recycling in the European Union Countries

PONE-D-23-29738R1

Dear Dr. Yilmaz,

We’re pleased to inform you that your manuscript has been judged scientifically suitable for publication and will be formally accepted for publication once it meets all outstanding technical requirements.

Kind regards,

Magdalena Radulescu

Academic Editor

PLOS ONE

Additional Editor Comments (optional):

Reviewers' comments:

Reviewer's Responses to Questions

**Comments to the Author**

1. If the authors have adequately addressed your comments raised in a previous round of review and you feel that this manuscript is now acceptable for publication, you may indicate that here to bypass the “Comments to the Author” section, enter your conflict of interest statement in the “Confidential to Editor” section, and submit your "Accept" recommendation.

Reviewer #1: All comments have been addressed

Reviewer #2: All comments have been addressed

2. Is the manuscript technically sound, and do the data support the conclusions?

Reviewer #1: Yes

Reviewer #2: Yes

3. Has the statistical analysis been performed appropriately and rigorously? 

Reviewer #1: Yes

Reviewer #2: Yes

4. Have the authors made all data underlying the findings in their manuscript fully available?

Reviewer #1: Yes

Reviewer #2: Yes

5. Is the manuscript presented in an intelligible fashion and written in standard English?

Reviewer #1: Yes

Reviewer #2: Yes

6. Review Comments to the Author

Reviewer #1: (No Response)

Reviewer #2: The authors have clarified their interpretations and applications by clarifying the limitations of certain methods, adding a helpful conceptual model, and improving the discussion. The authors employ clear and concise language throughout the manuscript, which enhances its accessibility to a broad readership.

7. PLOS authors have the option to publish the peer review history of their article (what does this mean?). If published, this will include your full peer review and any attached files.

Reviewer #1: No

Reviewer #2: No

---

## [Editor Report · Acceptance letter]

9 Feb 2024

PONE-D-23-29738R1 

PLOS ONE

Dear Dr. Yilmaz, 

I'm pleased to inform you that your manuscript has been deemed suitable for publication in PLOS ONE. Congratulations! Your manuscript is now being handed over to our production team.

Kind regards, 

on behalf of

Dr. Magdalena Radulescu 

Academic Editor

PLOS ONE